# 3D Bioprinting for Vascularization

**DOI:** 10.3390/bioengineering10050606

**Published:** 2023-05-18

**Authors:** Amatullah Mir, Eugenia Lee, Wesley Shih, Sarah Koljaka, Anya Wang, Caitlin Jorgensen, Riley Hurr, Amartya Dave, Krupa Sudheendra, Narutoshi Hibino

**Affiliations:** 1Section of Cardiac Surgery, Department of Surgery, University of Chicago, 5841 S. Maryland Ave., Chicago, IL 60637, USA; 2Pediatric Cardiac Surgery, Advocate Children’s Hospital, 4440 W 95th St. Oak Lawn, IL 60453, USA

**Keywords:** bioprinting, tissue engineering, vasculature, 3D modeling, scaffolds, spheroids, 3D printing, organ-on-a-chip, vascularization, hard tissue, stem cells, tissue engineering, biomechanics, biomanufacturing, bone, skin, oral diseases, 3D-printed sensors, bioinks, 3D tissues, soft tissue

## Abstract

In the world of clinic treatments, 3D-printed tissue constructs have emerged as a less invasive treatment method for various ailments. Printing processes, scaffold and scaffold free materials, cells used, and imaging for analysis are all factors that must be observed in order to develop successful 3D tissue constructs for clinical applications. However, current research in 3D bioprinting model development lacks diverse methods of successful vascularization as a result of issues with scaling, size, and variations in printing method. This study analyzes the methods of printing, bioinks used, and analysis techniques in 3D bioprinting for vascularization. These methods are discussed and evaluated to determine the most optimal strategies of 3D bioprinting for successful vascularization. Integrating stem and endothelial cells in prints, selecting the type of bioink according to its physical properties, and choosing a printing method according to physical properties of the desired printed tissue are steps that will aid in the successful development of a bioprinted tissue and its vascularization.

## 1. Introduction

The vascular system is adaptable—the various layers in blood vessels continually interact in efforts to remodel and maintain adequate perfusion to their respective tissues. The body not only develops blood vessels at the embryonic level, but continues to remodel and develop vasculature in adults well past embryogenesis. These processes are called vasculogenesis and angiogenesis, respectively.

Vasculogenesis is the de novo formation of blood vessels in the embryo. This occurs primarily through interactions between vascular progenitor cells, endothelial progenitor cells, and various growth factors during embryonic development. Specifically, mesodermal cells undergo receptor-mediated hemangioblast formation and later differentiate into endothelial cells in response to vascular endothelial growth factor (VEGF). Once endothelial cells have been formed, they coalesce to form hollow blood vessels. In the context of tissue engineering, research on endothelial progenitor cells (EPCs) involved in vasculogenesis has revealed the possibility of growing EPCs ex vivo to be later transplanted to allow for the neovascularization of tissue [1]. The usage of various stimulating factors, such as stromal-derived factor 1, to further stimulate vasculogenesis through EPC recruitment has also been of interest [2].

Angiogenesis, on the other hand, is the process by which new blood vessels are derived from the endothelial layer of a pre-existing vessel. In the adult, angiogenesis is the primary process by which vasculature is remodeled and built upon. Typically, a lack of nutrients or oxygen in a specific area triggers endothelial cells to break from a stable position and branch out towards the site, beginning the process of angiogenesis [3]. The remodeling of vasculature through angiogenesis is dependent on homeostatic mechanisms. The remodeling of vasculature is a long-term, strategic process by which endothelial cells interact with circulatory bulk to determine the appropriate modifications. Often, these interactions are with growth factors that stimulate endothelial cell proliferation, migration, and differentiation. VEGF, basic fibroblast growth factor (bFGF), and platelet-derived growth factor (PDGF or CD31) are three of the most prominent growth factors in angiogenesis [1].

Angiogenesis-promoting strategies have been of interest in tissue engineering research, which is primarily focused on the angiogenesis of capillaries within implanted constructs [4], temporal delivery strategies through degradation-dependent or trigger-specific release [5], and spatial control of angiogenesis through 3D bioprinting [6], among others.

Many cardiovascular pathologies have the potential to impede, deteriorate, or harm the body’s vasculature. For example, coronary artery diseases, stroke, atherosclerosis, aortic aneurysms, and others, can all cause vascular damage and can thoroughly diminish a person’s quality of life. Although drug and surgical treatments for vascular diseases exist and have been shown to improve patient conditions, several concerning flaws exist for traditional therapies such as drug durability and efficacy, post-surgical complications, immune rejection, and lingering symptoms [1,3]. Continually, although the development of stem cell research has sparked a new area of research in minimally invasive stem cell integration, they continue to have low viability and weak host cell integration.

Treating vascular diseases, let alone other pathologies and traumas, is difficult, and by no means is the scientific community ready with perfect solutions. Nevertheless, the advent of tissue engineering and 3D bioprinting has inspired new research that has the potential to push the boundaries of the current standard of care and regenerative treatment for disease and trauma. Vascularization plays a pivotal role in achieving sizeable and complex in vitro models, because tissue models with a thickness larger than 400 μm require vasculature to ensure cell viability. In addition, the recapitulation of blood vessel is important to improve the similarity of tissue models by enriching the microenvironment, and it is important for observing crosstalk between blood vessels and organs [7]. In this respect, 3D bioprinting techniques facilitate the generation of vascularized in vitro tissue models, because endothelial cells (ECs) can be positioned to form lumen structure effortlessly.

3D modeling and bioprinting has arisen as a means to regenerate tissue and organs such as bone, skin, and heart tissue printing. For instance, to promote bone regeneration, Keriquel et al. directly printed mesenchymal stromal cells in situ into a mouse cranial defect [8]. Additionally, Maiullari et al. developed a 3D-bioprinted construct using extrusion-based printing and mixed hydrogels and incorporated HUVECs and iPSC-CMs into the print [9]. There are several factors that must be evaluated in fabricating these 3D-bioprinted constructs, such as perfusability, conductive and contractile properties, and biocompatibility.

This manuscript discusses and analyzes recent developments in tissue engineering and 3D bioprinting to better characterize the current state of the field and discuss the potential for future work. We cover the latest advances in the process of 3D bioprinting. Specifically, this review focuses on 3D-bioprinted tissue model fabrication, including descriptions of the various cells used in the 3D bioprinting of various models, the selection of bioinks available for bioprinting, and fabrication methods, such as stereolithography, laser-assisted printing, inkjet, and microextrusion techniques. Moreover, we introduce various examples of implantation and evaluation methods of bioprinted models in vivo and in vitro. Finally, we discuss limitations and future trajectories of 3D bioprinting for various scale models.

## 2. Importance of Vasculature in 3D Models

Vasculature is primarily important for facilitating necessary oxygen and nutrient delivery, waste removal, and the circulation of immune cells [10]. Furthermore, as a critical part of functional tissue, vasculature is key to accurate, effective, and clinically relevant 3D-printed models [1,10]. Without successful vascularization, the host tissue would not be able to survive long enough to be useful in its applications. When fabricated tissue is implanted in vivo, vasculature allows implanted tissue to fully connect with existing tissue.

Precise vascularized 3D-biofabricated models are also valuable for medical research because they reduce the costs of drug testing and vascular disease modeling [11,12], and they are needed for high-demand research, such as tissue repair and regeneration, long-term tissue implantation, cancer research, and 3D cell culture. Specifically, researchers such as Kolesky et al. and Yu et al. [11,13] explained how organ-on-chip technology works. In this method, a perfused vascularized model provides accuracy while increasing viability and survival length significantly when compared to a non-perfused vascularized model. Therefore, 3D-vascularized models are important for multiple research purposes and provide an accurate way to mimic in vivo tissue.

There are several factors involved in the fabrication of vasculature (Figure 1).

### 2.1. Cells Used

To create vasculature, particular cells are needed for the bioprinting process (Table 1). Mesenchymal stem cells (MSCs) are advantageous in their multipotency and ease of access [3]. Cardiac stem cells (CSCs), which are able to differentiate into several relevant cell types, are also important for proliferation in vitro [3]. CSCs can also be combined with a pre-fabricated biomimetic microvessel to create a vascularized cardiac patch [14]. Embryonic stem cells (ESCs) are often chosen for their unlimited self-renewal properties. However, the use of ESCs is hindered by their tendency regarding tumorigenicity and immunogenicity, as well as ethical concerns arising from their attainment. Induced pluripotent stem cells (iPSCs) can be obtained relatively easily and are pluripotent, thereby leading to the possibility of a higher quantity of cells being acquired. However, their use is restricted by their low efficiency and high tumorigenic tendencies [3]. Human cardiac-derived cardiomyocyte progenitor cells (hCMPCs) are able to undergo cardiomyocyte differentiation and amplify in vitro [15]. In addition to these cell types used, cocultures of several cell types, including non-cardiomyocytes (such as endothelial cells and fibroblasts), have been shown to improve the structural formation of cardiac vascular tissue [3].

### 2.2. Use of Additives

To develop vascular bioprinted structures, many studies have referenced the use of additives such as growth factors [10,16,17,18]. Additives, such as vascular endothelial growth factor (VEGF), basic fibroblast growth factor (bFGF) and hepatocyte growth factor (HGF), promote neovascularization, which activates endothelial cells to stimulate cell assembly and support vessel formation and maturation [15]. Furthermore, in vivo VEGF-exposed collagen disks have demonstrated vascularization after 10 days compared to vascularization in non-VEGF-exposed cases [18]. In particular, it has been shown that VEGF, when released slowly, supports greater vessel formation and sees more CD31 expression than in cases with more rapid release [10].

## 3. Large-Diameter Vascular 3D-Bioprinting Techniques

Most current 3D bioprinting studies investigate the creation of small-diameter vasculature (<6 mm), because there are already alternative solutions for repairing large-diameter vascular channels that do not require 3D bioprinting techniques, such as the use of synthetic vascular grafts and autologous grafts [19,20,21]. There is, however, one study by Kucukgul et al. that aims for a scaffold-free approach for blood vessel fabrication by scanning a section of aorta and developing a 3D model using an algorithm. The study used 3D extrusion methods to generate cylindrical aggregates layer-by-layer and hydrogel support structures by using cellular aggregate and hydrogel as bioinks [22].

## 4. Small-Diameter Vascular 3D-Bioprinting Techniques

### Scaffold-Based Printing Methods

Printing scaffold-based models commonly use inkjet, extrusion-based, laser-assisted, and stereolithographic printing (Figure 2). Inkjet bioprinting uses forces to expel successive drops of bioink onto a substrate [23,24]. Because it can create complex structures and has high cell viability, it is often used with naturally occurring bioinks to create vasculature (Table 2).

Microextrusion bioprinting has been used to print a perfusable vascular network with lumen diameters as small as 100 μm [25]. It is the most commonly referenced technique for small vasculature printing and is used with some naturally occurring bioinks and with all synthetic bioinks for creating scaffolds. Although it can also directly print cells, doing so requires a larger nozzle size [25].

Laser-assisted bioprinting directs laser pulses through a ribbon-containing bioink to create the desired structure [24]. Since laser-assisted bioprinting is nozzle-free, has high cell-viability, and is as precise as one cell per droplet, it can use more viscous materials. It has already been used to create 5 μm diameter channels that have been successfully seeded. Using this method for large vessels would be inefficient [25]. Keriquel et al. successfully used laser-assisted bioprinting to directly print mesenchymal stromal cells associated with collagen and nano-hydroxyapatite to promote bone regeneration in a calvaria defect model in mice (Figure 3) [8].

Stereolithography (SLA) bioprinting uses the repeated photopolymerization of successive bioink layers to create a solid polymer of the desired design [24,26]. Common biomaterials for this method include polypropylene fumarate with photocrosslinkable bonds and polyethylene glycol acrylate [27]. Because stereolithography involves exposure to UV light, which can trigger cell-death, it is primarily used with synthetic materials to print scaffolds rather than to print cells directly. With current technology, SLA bioprinting is limited to a print resolution of about 300 μm, thus preventing it from printing even smaller diameter vasculature [25].

Cylindrical blood vessels with an internal diameter of nearly 1 mm and wall thickness of 0.3 mm have been produced through extrusion-based bioprinting with vascular endothelial cells [28]. The bioprinter in this study was modified to include a coaxial needle extrusion system, with two syringes containing differing bioinks, and two different cell layers (the vessel walls were seeded with vascular smooth muscle cells, and there was also an additional inner layer lining this wall consisting of vascular endothelial cells). Another coaxial needle extrusion method was shown to produce vascular channels with an average conduit diameter of 1449 μm and an average lumen diameter of 990 μm [29]. Branched conduits in this study were also formed by inserting smaller branch conduits into the wall of a larger branch conduit using microsurgery scissors and were sealed with alginate solution. Microvasculature was also produced through thermal inkjet printing [3]. This study printed fibrin fiber channels and was able to create fibers with an average diameter of 98 μm, and the endothelial cells that were printed lined the fibrin scaffolds to form channels after 21 days of culturing.

## 5. Scaffold-Based 3D-Bioprinting Materials

The following materials are classified as scaffold-based materials for 3D-bioprinting (Table 3).

### 5.1. Hydrogels

Hydrogels are effective for 3D vascularization due to their biocompatibility, high permeability, and their biodegradability, as well as the availability of binding sites for cell adhesion and growth factors [36,37,38]. Hydrogel bioinks can be developed as liquid precursors or as solid gel materials that allow hydrogels to be both structurally supportive and elastic, which is beneficial for withstanding pressures from blood flow [39,40]. They are also porous, which allows nutrients, waste, and growth factors to diffuse through the hydrogel, as well as allows for greater vascular viability and the possibility of cell signaling. While hydrogels maintain cell viability in this way, they are not a particularly viable option for stable prints. Many hydrogel bioinks have been developed with characteristics to resolve this issue in extrusion-based printing. One instance of a modified hydrogel bioink is the generated granular hydrogel bioinks created by Cheng et al., which contained microscaled hydrogel particles that were each composed of a crosslinked polymer network and water [39].

For vascularization, most hydrogels are composed of some combination of methacrylated gelatin hydrogel (GelMA) and natural polymers [38,40]. However, other types of hydrogels have also been shown to be beneficial for vascularization. Notably, hyaluronic acid (HA) hydrogels have been shown to be a suitable 3D microenvironment for tumor vascularization, and it has supported vacuole and lumen formation, vascular branching, and sprouting [41]. Poly(ethylene glycol) (PEG)-based hydrogels are also used to create vasculature and are often used with other bioactive polymers [41]. Matrigel is also another type of hydrogel that has been used with multiple types of vascular cells, because it has good cytocompatibility, cell adhesion, and can have varying physical properties [41]. Overall, the hydrogel vascularization extent has been found to differ based on hydrogel identity and crosslink density. Specifically, a lower crosslink density better supports vascular network formation [42].

### 5.2. Natural Polymers

An ideal scaffold would be the naturally decellularized extracellular matrix (dECM) because it provides the natural cell growth environment with proteins, growth factors, and other molecules that are important for cell function [29,43,44]. Currently, dECMs from the cartilage, liver, heart, brain, and other organs have been used to create 3D scaffolds [45]. However, the use of the dECM presents a number of challenges, including mechanical weakness and inconsistent production [45]. Inflammatory responses can result when decellularization methods are not thorough, and dECMs from different donors can contain different microenvironments [46,47]. Recently, Lee et al. used a combination of dECMs and elastic polymers to regenerate vascularized adipose tissue [48]. They found that the incorporation of the dECM facilitated neovascularization and tissue formation.

Due to the difficulties of dECM production, researchers have utilized alternatives such as the use of the natural polymers and proteins within the dECM to mimic the extracellular matrix [29,43,49]. To create the microvasculature, collagen has been consistently incorporated due to its ability to improve cell viability and support the vascular morphogenesis and network formation of multiple types of vascular cells. For example, an experiment performed by Huling et al. yielded a highly biomimetic hollow collagen vascular scaffold using segments of a rat’s left kidneys as the model. The collagen-dipped segments were dipped with collagen, dried, and soaked in a crosslinking solution to allow crosslinking to proceed. After acetone was used to remove the segments and the collagen vascular scaffold remained, this maintained hollow branching structures that could allow perfusion similar to native blood vessels. Huling et al. determined that it was possible to create a collagen vascular scaffold that copies the native vascular network in a healthy rat kidney by using crosslinking techniques [50].

Other natural polymers and proteins such as elastin, fibrinogen, sodium alginate, and gelatin have also been used to enhance vascularization. Many studies use these polymers and proteins in combination with GelMA hydrogel as an ink for the 3D print. For example, Lee et al. investigated how a bioink composed of GelMA and methacryloyl-substituted recombinant human tropoelastin (MeTro) could create a vascularized cardiac tissue construct [51]. Tropoelastin, an elastin precursor, was used because it has been shown to support endothelial cell recruitment, migration, and angiogenesis. As a result, Lee et al. were able to fabricate an entirely 3D-printed artificial tissue that demonstrated endothelium barrier function and spontaneous cardiac cell beating, as well as minimal inflammatory responses, when implanted in rats [51].

Many studies also utilize a combination of alginate and gelatin hydrogels to create vasulature. Gentile et al. demonstrated cardiomyocyte contractility and endothelial cell self-organization after 14–18 days in a culture using a mix of alginate and gelatin [52] (Figure 4). Using alginate-based hydrogels, Luo et al. demonstrated endothelial monolayer formation and dense HUVEC connection. They 3D printed a concentrated alginate/gelatin combination and soaked the scaffolds in crosslinking solutions. Only the outside parts of the scaffold became crosslinked, which created a hollow channel from the uncrosslinked portions [53].

Fibrinogen has also been used in combination with gelatin to create small-diameter constructs with the desired shear-thinning property for rotary bioprinting [54]. The fibrinogen/gelatin bioink was mixed with low passage primary neonatal human dermal fibroblasts (HDF-n) during the printing process and crosslinked using thrombin. The results showed increased collagen deposition and increased mechanical strength within 2 months of establishing the cultures [54].

### 5.3. Synthetic Polymers

Synthetic polymers offer significant variability, because their mechanical properties can be easily controlled, thus making them a flexible, tunable means of scaffold-building (Figure 5). Synthetic polymer-based scaffolds may be processed under mild conditions at a relatively low cost. However, they lack the vital biological cues responsible for enabling certain cellular responses. Specific types of synthetic polymers include the following, which are shown below:

Polylactic acid (PLA), which is a degradable aliphatic polyester that has properties of biocompatibility, degradability, and printing ability that make it a prominent polymeric bioink, is typically used in generating filaments that are used in musculoskeletal tissue engineering. However, due to the brittle nature of PLA, PLA is combined with other low-cost ceramic materials, such as calcium phosphate, in order to form scaffolds for improved bone strength and reduced acid generation [55].

Polyether ether ketone (PEEK), which is a nonbiodegradable polymer, contains properties of low heat conductivity, high biocompatibility, heat resistance, and chemical stability. As a result of its crystalline structure, it has strength and elasticity that make it useful in FDM and SLS technologies, and it is specifically involved in craniofacial implants and bone replacement as a result of its physical and chemical properties [55].

Poly-D,L-Lactic Acid (PDLLA) is a synthetic polymer with an amorphous structure, hydrophobic properties, and that contains mechanical properties that make it helpful in forming biocompatible scaffolds. Its application is found in orthopedic rehabilitation and the tissue engineering of resorbable devices. Because PDLLA is a hydrophobic polymer, it lengthens the decaying due to the lack of water diffusion in its matrix [55].

Another synthetic polymer is acrylonitrile butadiene styrene (ABS), which is a triblock copolymer with a low melting point, tensile strength, and toughness. Because ABS is made of three different monomers, these provide a number of beneficial properties such as heat endurance, impact strength, and rigidity. These properties make ABS useful in cartilage engineering technologies and selective laser sintering printing [55].

Polyethylene glycol (PEG) is a hydrophilic polymer used in tissue engineering scaffold formation because of its enhanced biocompatibility. PEG is, however, non-biodegradable and has poor mechanical strength; PEG can be degraded hydrolytically and enzymatically [55].

Poly-glycolic acid (PGA) is a main synthetic polymer in 3D scaffolding because of its versatility, biocompatibility, and other biological features. PGA is used in bone internal fixation devices and resorbable suture preparation. Through the hydrolysis of ester bonds, PGA’s seeding density may be increased [55].

Poly caprolactone (PCL) is advantageous for its stiffness, biocompatibility, and degradability, as well as for its cost effectiveness and toleration of stability. The stability, and accompanying features of PCL leads to the development of secondary obstacles. Additionally, the higher hydrophobicity of PCL results in low reduction and low bioactivity [55].

Polybutylene terephthalate (PBT), which is a biocompatible thermoplastic polyester, contains high elasticity, appropriate strength and toughness, and an ease of processing. It is typically employed in FDM printing and in the biomedical field for in vivo and in vitro biocompatibility. Specifically, it is used in printing bone scaffolds, tissue regeneration, and as filler in orthopedic surgery, while PBT has typical advantages similar to PCL or PLA. Its high melting point limits its applicability [55].

Polyurethane (PU) is a biodegradable elastomer with high biocompatibility and mechanical strength. It may be differentiated by the use of solvent, either in water-based PU or solvent-based PU. The characteristics of 3D bioprinting with PU include increased printing resolution and good cytocompatibility. PU is the optimal synthetic polymer for cartilage tissue engineering, bone fabrication, and the construction of muscle and nerve scaffolds. Its elastomeric properties make it the optimal choice for muscle generation [55].

Poly-vinyl alcohol (PVA) is a synthesized polymer in the presence of vinyl alcohol and acetate, both of which, as monomers, contribute to PVAs being biocompatible, biodegradable, bioinert, and sem-crystalline in nature. PVA is endurable to more intense environments and enables the efficient intracellular diffusion of oxygen and nutrients [55].

Polylactic-co-glycolic acid (PLGA) is a polymer with a reliable biodegradable nature and cytocompatibility properties. PLGA is also used in bone regeneration and tissue restoration systems. However, its hydrophobic features and linear structures lead to poor mechanical stiffness, high degradation rate, and downgrade its status as a scaffold material. As a result, PLGA is often paired with PCL for success in scaffolding [55].

## 6. Scaffold-Free Printing Materials

The creation of spheroids, a collection of cells, via hanging drop, spinner flask, or the Kenzan method is an increasingly common way to avoid scaffolding [10]. The Kenzan method uses a bioprinter to precisely assemble and print spheroids, thereby letting them fuse together to form cell aggregates and create their own extracellular matrix (ECM) [56]. While spheroids express strong biomimicry of native structures and mature constructs, a large number of spheroids are required [10]. The spheroids of endothelial cells have been used to create vasculature by assembling them in the structure of a vessel [2,57,58] or by loading 3D-cultured spheroids into a hydrogel structure. The latter method allows cells to self-assemble into clumps and develop intercellular linkages and vascular networks prior to being placed in the tissue structure, thereby mimicking the developmental trajectory of tissue formation in humans [59]. Daly et al. presented a method of 3D bioprinting with the infusion of spheroids within self-healing hydrogels. These iPSC-derived cardiac microtissue models were bioprinted with spatially controlled cardiomyocyte and fibroblast cell ratios in order to mimic existing scarred cardiac tissues that develop as a result of myocardial infarction [60] (Figure 6).

### 6.1. Spheroids

The creation of spheroids, which are a collection of cells, via hanging drop, spinner flask, or the Kenzan method is an increasingly common way to avoid scaffolding [10]. The Kenzan method uses a bioprinter to precisely assemble and print spheroids, thereby letting them fuse together to form cell aggregates and create their own extracellular matrix (ECM) [56]. While spheroids express a strong biomimicry of native structures and mature constructs, a large number of spheroids are required [10]. Spheroids of endothelial cells have been used to create vasculature by assembling them in the structure of a vessel [2,57,58] or by loading 3D-cultured spheroids into a hydrogel structure. The latter method allows cells to self-assemble into clumps and develop intercellular linkages and vascular networks prior to being placed in the tissue structure, thereby mimicking the developmental trajectory of tissue formation in humans [59].

### 6.2. Multi-Material Bioprinting

Multi-material bioprinting allows for the simultaneous or consecutive deposition of bioink materials and combined component materials. Several of such printing methods are currently in development, one of which is coextrusion printing. Coextrusion allows for simultaneous deposition by laying down a mixture of bioprinting materials, such as cells, bioinks, or growth factors [61]. This method is useful for positioning different cell types in a controlled manner to promote cell–cell interactions or for immediate cross-linking via a fugitive ink to strengthen the hydrogel scaffold. While coextrusion bioprinting involves material placement through one nozzle, other techniques use multiple nozzles to achieve a particular material deposition pattern [61]. Most of these multi-jet techniques allow for the more specific placement of single materials, which are each loaded into their own cartridges though the intricate needle multi-jet (INMJ) method, which allows for the concentric placement of biomaterials through overlaid needles of different gauges.

Jia et al.’s coaxial system allowed for the printing of a 3D structure in hollow tubes of bioink that were instantly stabilized by a calcium solution upon deposition. This created a strong and controlled matrix for cell deposition, which was demonstrated by the successful coculturing of HUVECs and MSCs in the mold [62]. The ink used was a combination of GelMA, sodium alginate, and 4-arm poly(ethylene glycol)-tetra-acrylate (PEGTA). Coaxial printing has also recently been used in combination with a vascular tissue-derived dECM bath [63]. The researchers created a triple-layered artery equivalent with a diameter of 600 μm that was able to successfully generate functional tissue [63].

In addition to pairing cellular materials with other components such as growth factors or bioinks, multi-material printing also aims to integrate a cell type of interest with vasculature and stroma to better recapitulate the native environment. Kolesky et al. [13] used a variety of printing techniques to create a cross-linked gelatin–fibrin skeletal matrix, which both retained structural integrity while being biocompatible, to create a vascular network to support thick bioprinted tissue that was >1 cm in width. The matrix was perfused with human umbilical vascular endothelial cells (HUVECs) to mimic the vascular lining, and the matrix space was filled with a human neonatal dermal fibroblast (hNDF) extracellular matrix and human MSCs as tissue that was continually perfused with media and growth factors. This intricate coculturing resulted in a complex, diverse tissue sample with evidence of proliferation, cell–cell interaction, and osteogenic differentiation, thereby mimicking natural development of the tissue in situ (Figure 7).

## 7. Analysis Techniques and Clinical Applications

### 7.1. In Vitro and In Vivo Techniques

In vitro techniques that are used to analyze vascularization include 3D endothelial cell culture or coculture, which can self-assemble tubule networks, or, alternatively, endothelial cell cocultures on beads embedded within an extracellular matrix, which produce quantifiable tubes [64]. Additionally, microfluidic models consisting of biologically linked hydrogels can allow for the analysis of precise geometries in blood vessel formation in particular [65]. In vivo analysis techniques include histological approaches, but it is limited in assessing the three-dimensional architecture and perfusion of vascularization. To receive a more quantitative result for assessing three-dimensional architecture in vivo, microfill perfusion can be performed on the system, and a cast can be made of the vasculature for analysis purposes [64].

### 7.2. Imaging

Analyzing through imaging involves several factors, including contrast, depth penetration, resolution, and energy tolerance. These factors are dependent on each other, and prioritizing one comes at the expense of another [66]. For the imaging of large tissue constructs, depth penetration is most important; selective plane illumination Microscopy (SPIM) is effective for depth penetration as well as image resolution. However, SPIM is limited in its reliance on contrast agents and is, therefore, not always optimal for analyzing in vitro tissues. Photoacoustic imaging is another imaging method, which is non-invasive and focuses on vascular formation to produce ultrasonic signals. In contrast to SPIM, this method is not dependent on the use of contrast agents. For the imaging of vascular flow rates (as opposed to vascular morphology), two notable methods are laser doppler scanning and laser speckle contrast imaging (LSCI). Laser Doppler scanning involves the observation of light being scattered across blood cells, and the corresponding Doppler frequency shift from this scattering is used to determine flow rates. For LSCI, there is also optical scattering, but instead of Doppler frequencies, random interference patterns and their phase shifts are analyzed to determine the flow rate [66]. Certain tissues have different ideal flow rates, so by analyzing the fabricated flow rates, researchers are able to determine the effectiveness of their systems.

### 7.3. Clinical Applications

To implement scaffold-based tissue engineering for vascularization in a clinical setting, the scaffold properties must be verified in connection to clinical cases [49]. Imaging techniques such as microCT scans and MRIs can be performed to evaluate the vascular configuration, while the connectivity of the vascular structures can be evaluated through perfusion. Vascular mechanical properties can be assessed with static and hydrodynamic loading, while biocompatibility can be evaluated through fluorescent stainings, flow cytometry, or MTT assays. The degradability of structures can be verified through scaffold weight measurements over time. Additionally, scaffolds need to be cultured with cell seeing or loading. In this phase, sterility must be maintained and cell morphology, proliferation rate, chemical composition, and viability must be analyzed to demonstrate clinical application feasibility.

## 8. Feasibility, Limitations, Challenges, and Future Perspectives

Although previous studies have attempted to use 3D bioprinting to create vascularized tissue patches, challenges and limitations still remain, thereby preventing this field from exponentially accelerating forwards. The difficulty in creating a sufficient blood vessel system generally stems from a difficulty in mimicking natural vascular architecture and rebuilding these networks in vitro.

Both unique and strategy-specific challenges arise within different tissue creation methods. Within the cell-based strategy, the growth of new vasculature tends to be too slow [16]. Additionally, neoangiogenesis is difficult to control due to the instability of released growth factors and the advanced techniques required for the coculturing of cells to mimic the in vivo environment [16]. Within the scaffold strategy, challenges arise from a difficulty in using biocompatible synthetic materials to create a porous small-lumen vessel that has a wall thickness of less than 5 μm [16]. Furthermore, 3D printing bioinks must be compatible for both the printing processes and the 3D cell cultures [49].

The main challenges stem from a difficulty in maintaining cell culture conditions and, ultimately, cell growth. Researchers have yet to uncover the optimal culture conditions that lead to vascular cell proliferation and migration, as well as how to regulate vascular cell growth [49]; research has not yet fully taken into consideration the levels of vascularization necessary for cell survival. Additionally, the general cell availability of resources remains a hurdle. When creating cell tissues, whether using 3D bioprinting or not, the immediate availability of O2 can be insufficient depending on the thickness of the cells; thus, it is difficult to maintain viable tissue thickness while simultaneously avoiding cell death, as the cells located at the core of the tissues may struggle to receive sufficient resources. On top of the aforementioned challenges, the translation of this research into clinics remains slow, as 3D bioprinting technology still struggles to match the rapid rates of clinical surgeries and the urgent needs of patients [49]; this slow printing speed leads to difficulty in preserving cell viability if tissues are to be printed on larger, potentially whole-organ, scales. Thus, the current tissue fabrication speeds prevent the creation of clinically-relevant amounts [1,49].

Although the advances within this field over the past few years have been profuse, there still exist limitations to the use of 3D bioprinting due to the lack of technological advances and an inability to produce rapid results.

Currently, we are unable to facilitate the vascularization and diffusion processes for thicker fabricated tissues. Typically, capillaries are located within 100 μm from most of the cells within our body, thus allowing for sufficient diffusion and cell survival [23]. Thus, for thicker-printed tissues, diffusion is necessary for cells to obtain resources. One potential way to overcome this issue, as proposed by Hutmacher et al, is the formation of an artificial vascular system, which can enhance waste removal and nutrient transportation [23].

Another limitation lies in the inability to mimic the complexity of real-life tissue due to the limited resolutions and makeup of created tissue structures. While 3D bioprinting has the theoretical ability to build structures with arbitrary geometric and material properties, the current printing system’s technology and bioink’s inability to create the arbitrary features prevent this field from experiencing the full extent of its capabilities [15]. Thus, it is difficult to create vascular scaffolds for complex tissues, such as the alveoli within one’s lungs [49]. Furthermore, our current hardware and printing materials prevent efficient vascularization. We lack printers that can easily accommodate multiple materials, so different cell types and ECM proteins still cannot be used to create these complex tissues [15].

In addition, current bioprinters lack a high-definition resolution that can help print small vessels and other fine features within tissues [66]. Finally, printing materials must be selected based on their biocompatibility, crosslinking methods, and extrusion characteristics, which thus limit the number of available materials that can be used and the feasibility of tissue creation [66]. We currently also lack a system that can create structures across all vessel scales. There are 3D bioprinters that can produce structures on the nanoscale and the macroscale, but there is not a single system that can create tissues spanning the size range of structures found in vivo [15]. The human body’s vasculature has multiple dimensions, ranging from large arteries down to small capillaries. In order for successful clinical translation, bioprinting systems must be able to recapitulate this multiscalar nature, yet current technology struggles to print capillaries that are submicron-sized [1]. In addition to the limitations with current technology, other factors limiting the advancement of 3D bioprinting include more fundamental issues. Current bio-engineered prints are not close to 100% cardiac replication, either physically or chemically [67]. Additionally, most comprehensive comparisons need to be made between bioprinted cardiac tissues and natural cardiac tissues within the molecular level, thereby leading to a disconnect between this research’s showcase application [67]. Despite the current gap between bench research and clinical impacts of 3D-printed tissues, improvements in technology and techniques can lead to the smooth integration of created tissues with native tissue. New technologies such as 4D printing—which entails the involvement of structures that can be deformed under external stimuli such as temperature, pH, light, or chemicals—can lead to improved scaffold resolutions [68,69,70]. Additionally, the combination of different 3D printing approaches and materials can reduce the pitfalls of one printing technique and the materials used [27]. For example, combining the self-assembly of spheroids and 3D printing-based methods leads to Kenzan bioprinting that uses the stacking process of cell spheroids [49]. Bioprinting with multiple materials can allow individuals to adjust factors, such as growth factor concentration, degradation rate, and cell adhesion levels, within different locations of a printed tissue, thereby mimicking the diversity within in vivo tissues [23]. Finally, our current research with 3D bioprinting paves the way for different levels of analysis, such as studying epigenetic mechanisms to fast forward the transition into clinical applications [67].

The most recent advances in 3D bioprinting have several implications for the future of 3D bioprinting for vascularization. In recent years, there have been significant advancements in the field of 3D bioprinting for vascularization, with a focus on tissue organoids, organoids-on-a-chip, and in situ bioprinting. These techniques aim to produce biomimetic tissue constructs that closely resemble the natural vasculature and allow for the creation of complex multicellular structures that more closely mimic the cellular composition of native blood vessels, thereby enabling the formation of functional vascular networks.

Firstly, tissue organoids, or small three-dimensional structures that closely mimic the complexity and function of real organs, are a recent focus in 3D bioprinting techniques. These are often used as building blocks to create more complex structures and vasculature [71]. This allows for the creation of highly complex and precise structures that mimic various organs and tissues [72]. Specifically, various recent studies have tested the applications of liver organoids in generating liver-specific vasculature for use in liver disease studies and medication testing. Due to the gradual anoxia of the organoid over time, the future experimentation of organoids test the incorporation of biomaterials, nanotechnology, and other advanced protocols that may enable the long-term culture of a human organoid [73].

Organoids-on-a-chip is another recent focus in new 3D bioprinting techniques, which involves the incorporation of an organoid into microfluidic devices. This allows precise control and manipulation of the surrounding microenvironment [74]. This combination of organoids and microfluidics helps generate vascular structures that resemble the in vivo microenvironment, as it provides a more physiologically relevant environment. The technology used in organoids-on-a-chip is more precise and controls the deposition of cells and extracellular matrix materials, which have the potential to generate a customized vasculature dependent on the needs of the individual patient and the tissue desired [75]. While the use of the chip is novel, it is limited in its use of certain cell types. Future directions and experimentation with chips involve incorporating patient-specific cells in order to avoid immune rejection of the chip by the host [76].

In addition to organoids and their incorporation into microfluidic devices, in situ bioprinting is another recent advancement in 3D bioprinting for vasculature. This technique allows for the direct printing of biomaterials onto the desired tissue or organ, thereby bypassing the use of a premade construct. In situ bioprinting overcomes challenges associated with traditional 3D bioprinting methods, such as the difficulty in achieving sufficient vascularization for sustaining the printed construct. This advancement in bioprinting promotes the formation of blood vessels and general vascularization via potentially minimally invasive printing techniques. Future techniques in testing in situ bioprinting explore the manufacturing of handheld printers so that surgeons may print in situ on a patient’s tissue or organ [77].

Overall, these novel techniques in 3D bioprinting for vascularization result in a more biomimetic environment for the development of functional blood vessels and vasculature within bioprinted tissues. With further research and development, in situ and organoid bioprinting for vascularization is poised to significantly impact the field of bioprinting, thus opening up new opportunities for the fabrication of complex, functional, and clinically relevant tissue constructs.

## 9. Conclusions

In summary, bioprinting vasculature can take place through extrusion-based, inkjet, stereolithographic, photolithographic, and laser-based prints. These methods are used to generate scaffold or scaffold-free prints. Scaffold prints typically consist of hydrogels, natural polymers, and synthetic polymers, whereas scaffold-free prints are made of decellularized extracellular matrices, spheroids, alginate hydrogels, and protein-based inks. Multimaterial bioprints can also be produced. Bioprinting the vasculature is most successful depending on the cells and printing technique used. Analysis can be conducted through in vivo and in vitro techniques, as well as imaging. Bioprinting processes have been enhanced by computational modeling through the optimization of tissue parameters for bioprinting. While significant developments in 3D printing vasculature have come to light, challenges and limitations still remain. Cell culture conditions and their maintenance in prints present the main challenge, in addition to the vascularization of thick tissues, which has proven difficult to facilitate. Mirroring real-life tissue complexity and scales are further limitations in 3D printing vasculature. Exploring these limitations and looking to newer technologies such as 4D printing are critical steps for enhancing the 3D printing process and generating successful prints.

## Figures and Tables

**Figure 1 bioengineering-10-00606-f001:**
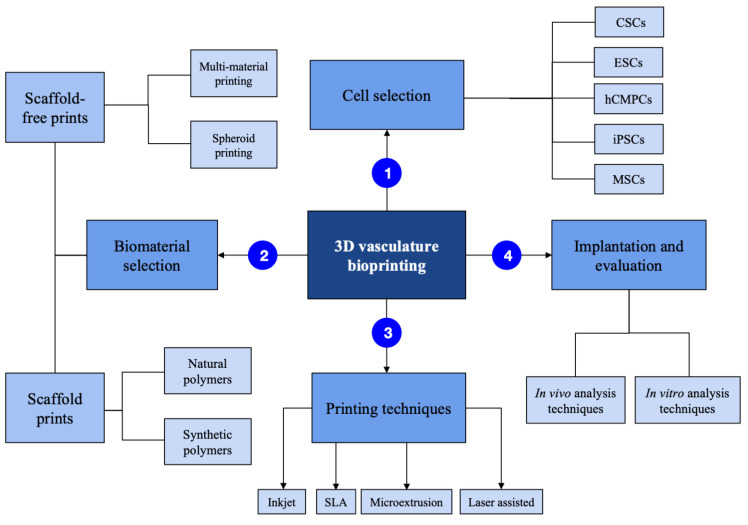
3-dimensional bioprinting process summarized.

**Figure 2 bioengineering-10-00606-f002:**
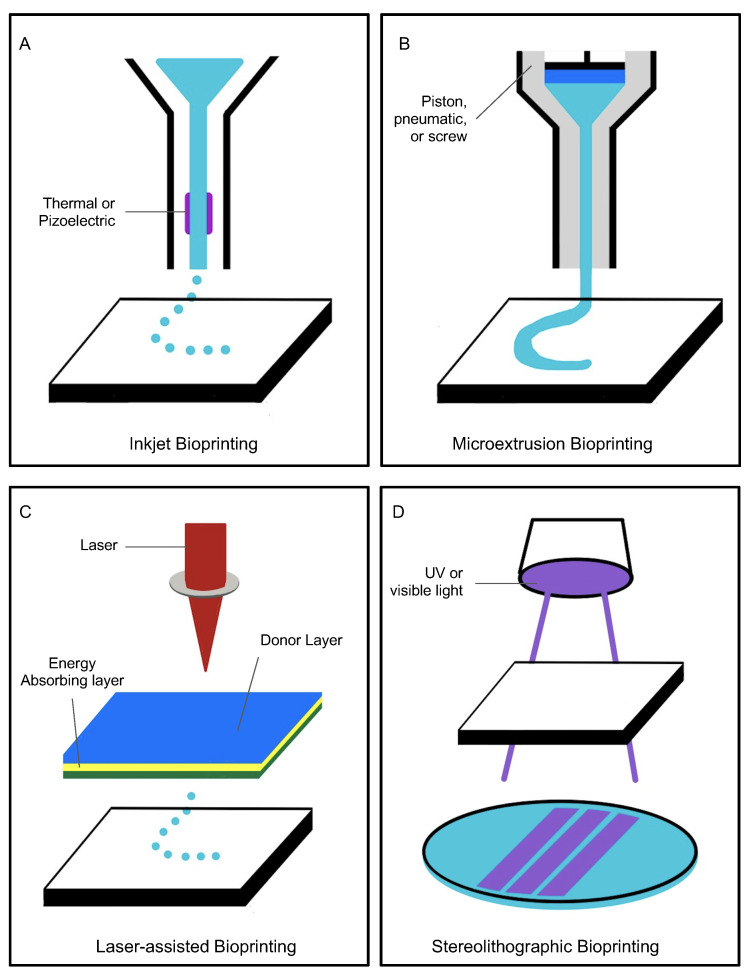
There are four methods of bioprinting: inkjet (**A**), microextrusion (**B**), laser-assisted (**C**), and stereolithographic (**D**).

**Figure 3 bioengineering-10-00606-f003:**
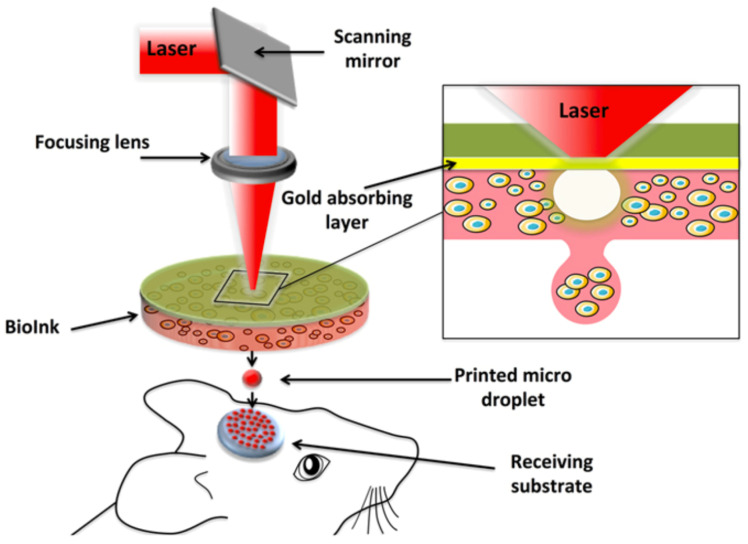
Schematic of the laser-assisted bioprinting (LAB) fabrication method and its direct printing onto a receiving substrate. Adapted from Keriquel et al. [8].

**Figure 4 bioengineering-10-00606-f004:**
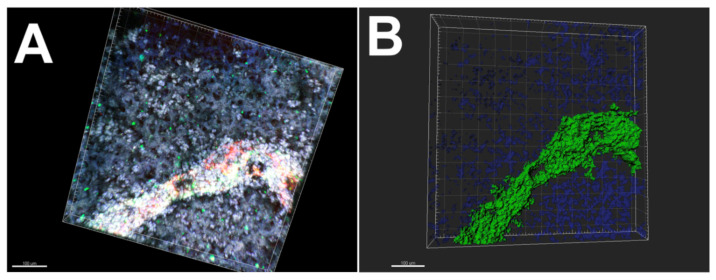
3D Imaris software analysis of 3D bioprinted patch stained with antibodies against CD31 (green) and Hoechst stain (blue). (**A**,**B**) 3D rendering of the endothelial structure of within the 3D bioprinted patch. Adapted from Gentile et al. [52].

**Figure 5 bioengineering-10-00606-f005:**
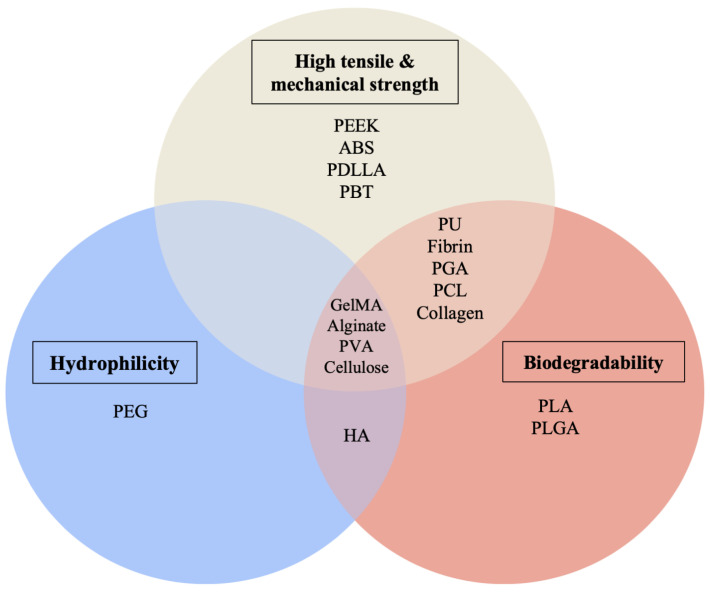
Natural and synthetic polymer properties, summarized.

**Figure 6 bioengineering-10-00606-f006:**
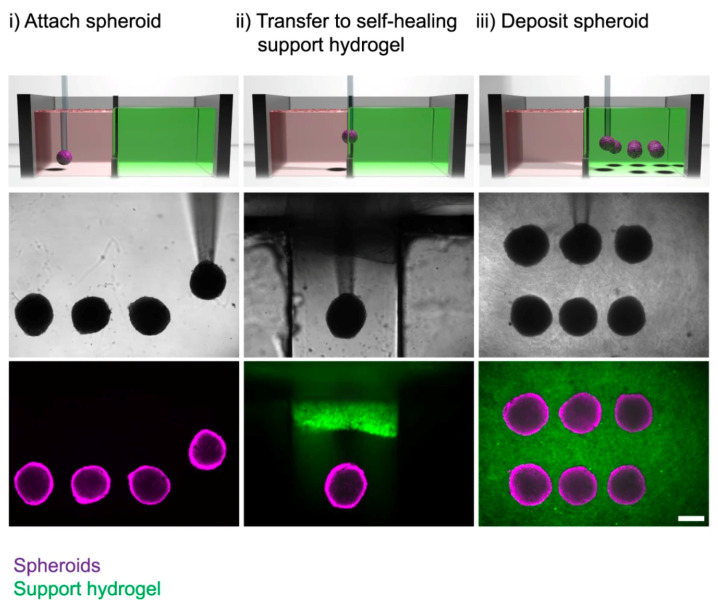
Spheroid aspiration, transfer, and deposition into a support hydrogel. Adapted from Daly et al. [60].

**Figure 7 bioengineering-10-00606-f007:**
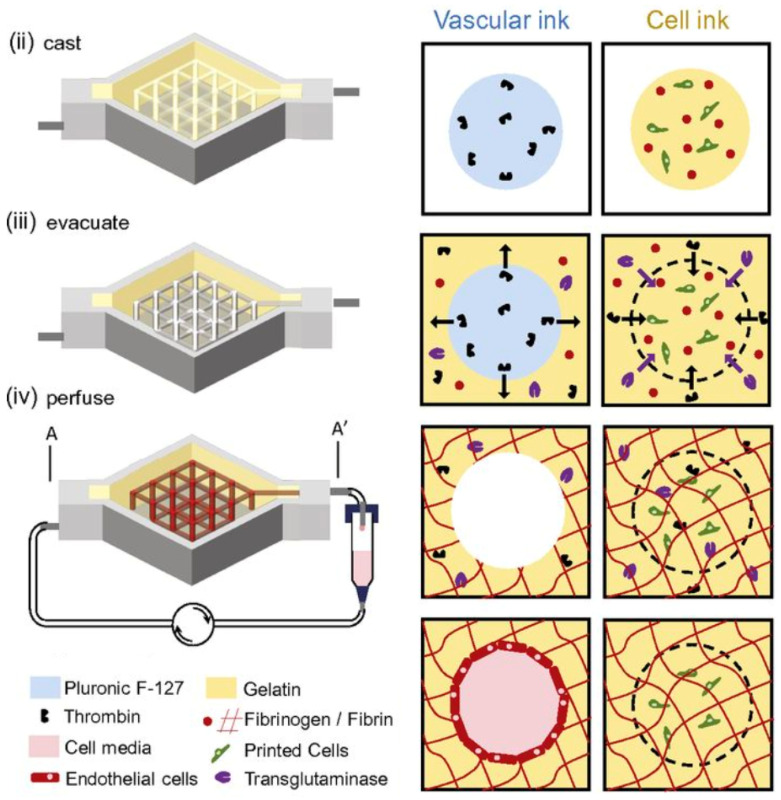
Pluronic, thrombin, and cell-laden inks, which contain gelatin, fibrinogen, and cells, are printed in a 3D perfusion chip, which is cast with ECM material. Thrombin and fibrinogen polymerize to form fibrin, and the system is perfused with an external pump. Adapted from Kolesky et al. [13].

**Table 1 bioengineering-10-00606-t001:** Cell types used in vasculature.

Cell Type:	Cardiac Stem Cells (CSCs)	Mesenchymal Stem Cells (MSCs)	Induced Pluripotent Stem Cells (iPSCs)	Embryonic Stem Cells (ESCs)	Human Cardiac-Derived Progenitor Cells (hCMPCs)
	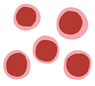	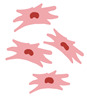	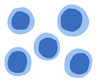	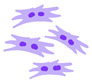	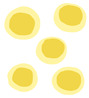
**Properties:**	Able to differentiate into various cell types for proliferation in vitro.	Easily accessed, advantageous in multipotency.	Easily accessed, pluripotency allows for higher cell attainment.	Contain unlimited self renewal properties.	Able to differentiate into cardiomyocytes, and amplify in vitro.
**Applications:**	Printing vascularized cardiac tissue	Bone and cartilage tissue	Heart, hepatic, bone, neural, and cartilage tissue	Embryoid body tissues	Printing vascularized cardiac tissue
**References:**	[6]	[6,7]	[6]	[7]	[7,8]

**Table 2 bioengineering-10-00606-t002:** Bioprinting techniques and their advantages and disadvantages in vascularization.

Bioprinting Technique	Advantages	Limitations	References
Inkjet bioprinting	Optimal for complex network structures. Cost-effective.	Heat and mechanical stress may impact cells incorporated in the printing process.	[16,17]
Extrusion-based	Optimal for printing small perfusable networks.	Not optimal for printing with cells, as larger nozzles are required for print.	[18]
Laser-assisted	Optimal for cell viability maintenance and increased precision printing. Can use viscous materials.	Inefficient for printing larger vessels.	[17,19]
Stereolithographic	Optimal for printing crosslinked solid polymers accurate to desired design.	Not optimal for printing directly with cells, as UV light exposure can trigger cell death. Not optimal for prints smaller than 300 μm.	[17,20,21]

**Table 3 bioengineering-10-00606-t003:** Characteristics of various bioinks that are used in scaffold-printing to develop 3D-bioprinted constructs.

Bioink	Category	Properties	Cell source	Reference
GelMA	Granular hydrogels	Provide a hydrated, biocompatible environment to the cells and maintain the shape of the printed materials. Largely aqueous and consist of part of a crosslinked polymer network	HUVECs, VSMC	[30]
Collagen	Natural polymers	Can also be combined with different polymers, such as alginate	HMVECs, HUVECs, SMCs	[31]
Alginate	Natural polymers	Ideal for bioprinting because of its ability to trap water and other molecules through capillary forces whilst allowing diffusion inside and out. Helps to form blood vessel tissues in vivo that are utilized as natural polymer scaffolds.	Cartilage progenitor cells	[32]
Hyaluronic acid	Natural polymers	Low mechanical properties and slow gelation behavior. Best for combining with various polymers in order to provide strong viability and other functional properties	HMVECs	[33]
Fibrin	Natural polymers	Can be combined with thrombin to form a fibrin hydrogel useful for its biocompatibility and biodegradation properties	HMVECs, HDF-n	[9]
Chitosan	Natural polymers	Promotes cell viability, is biodegradable, and maintains cell viability.	HUVECs	[34]
Cellulose	Natural polymers	Has excellent biocompatibility and maintains cell viability.	HSCs	[35]

## Data Availability

The data presented in this study are publicly available as found in the references below.

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
