# Peer review of "3D Bioprinting for Vascularization"

_bioengineering, 2023, doi:10.3390/bioengineering10050606_

Round 1

Reviewer 1 Report

Bioengineering-2166062

In this manuscript, the authors provide a comprehensive review regarding the 3D Bioprinting for Vascularization. It is eye-catching and exciting to a researcher in this field. However, some corrections and explanations must be done previous to publish the paper in Bioengineering. Some of my comments and questions on this manuscript are as follows:

  1. The manuscript contains huge textual documentation, which makes it very boring for the readers. I would suggest the authors add some of the figures or add new images to the single image, and schematic illustrations such as Classification of the vascular networks and materials used for bioprinting make it more interesting.
  2. Comparison of 3D bioprinting techniques used for fabricating vascular-like structure could be presented in the form of Table or in the main text
  3. I suggest that the author presents the schematic presentation of 3D bioprinting of in situ vascularized tissue engineered bone construction and application in repairing bone defects.
  4. I suggest that the author presents the schematic presentation 3D Bioprinting Process including CT scanning/MRI technique gets the 3D image of the damaged part for patients; CAD models; Bioink preparation; Bioprinting of the 3D constructs using 3D Bioplotter; Application of 3D bioprinted parts
  5. The author could provide information about the bioinks for vascular and vascularized tissue.
  6. The author could provide a summary of advantages and limitations of different bioprinting techniques, in the context of vascular tissue engineering. 
  7. Some references about bioinks may be useful for this Review Article: Materials 2020, 13 (18), 3980; Polymers 2022, 14 (11), 2238; Tissue Engineering Part A 27 (11-12), 679-702. 
  8. Surprisingly small references to Bioengineering in the literature despite the large relevant literature there. This should be improved. There are several important papers in recent literature.

Reviewer 2 Report

Despite the review, the contents and references are too poor to accept. First, the authors should mention the concept and importance of 3D models (not limited to vasculature). Why the 3D models are helpful and what biomaterials are used are essential. In the current version, the manuscript cannot be published. Taken together, the manuscript would be re-considered only when all the comments are responded.

1. Introduction

Too short. The authors must introduce the concept of 3D models with several tissues. Without the sentences, this review is not appropriate. The reviewer suggests the paper to be cited.

Review (for concept)

Bone: doi.org/10.1016/j.bioactmat.2017.10.001

Cancer: Cancers 202012(10), 2754

Skin: doi.org/10.3389/fbioe.2018.00154

2. Importance of …..

The reviewer thinks the importance is not enough, such as for drug permeability, drug resistance, fragility, and so on.

3. Bofore 5.1.

The authors should introduce the representative biomaterials for scaffolds.

The reviewer suggests the paper to be cited.

Alginate https://doi.org/10.1080/00914037.2018.1562924

Collagen  https://doi.org/10.1002/bip.20871

Gelatin https://doi.org/10.3390/molecules26226795

Chitosan  doi.org/10.1016/j.ijbiomac.2018.04.176

HA  https://doi.org/10.1016/j.carbpol.2012.10.028

Reviewer 3 Report

In this Manuscript entitled “3D Bioprinting for Vascularization”, the authors reviewed methods of printing, bio-inks, and analysis techniques in 3D bioprinting for vascularization. Additionally, they also evaluated to determine the most optimal strategies of 3D bioprinting for successful vascularization. Finally, this review incorporates recent advancements, which will allow the reader to understand the implications of bioprinting on 3D printing tissue engineering and future areas of research in developing new technologies.

Overall, the review article still has some shortcomings, which need to be addressed before possible publication in this journal.

Please find the attached annotated file to see my comments.

Lastly, I would like to say Bioengineering is a competitive journal, which published high-quality review articles related to 3D/4D bioprinting. Based on my comments, the recommendation is Major Revision.

Reviewer 4 Report

Excellent reviews on recent 3D bioprinting for vascularization have already been published (e.g. 10.3389/fbioe.2021.664188 (2021), 10.3389/fbioe.2021.685507 (2021), 10.1186/s13062-020-00273-4 (2020), 10.1016/j.matdes.2020.109398 (2021)). These reviews show many figures and cover the contents of the present review. In addition, Fig. 1 is the almost same of the image in the previous review (10.3389/fbioe.2021.664188). Therefore, the present review might not attract readers, although this topic is interesting. This reviewer is negative for acceptance.

Other comments

1. A space should be added between a number and a unit. For example, “6mm” should be “6 mm”.

2. “um” should be “µm”.

Round 2

Reviewer 2 Report

After all, the contents and references are too poor to accept.

I think this review is not suitable for publication.

Author Response

We thank the reviewer for their comment.

Reviewer 3 Report

The article is more improved than the previous one and can be accepted now.

Author Response

(The authors gave the same response as above.)
